# Effects of soil particles and convective transport on dispersion and aggregation of nanoplastics via small-angle neutron scattering (SANS) and ultra SANS (USANS)

**Anton F. Astner**[1], **Douglas G. Hayes**[1]*, **Sai Venkatesh Pingali**[2]*, **Hugh M. O'Neill**[2], **Kenneth C. Littrell**[2], **Barbara R. Evans**[2], **Volker S. Urban**[2]

**1** Biosystems Engineering and Soil Science, The University of Tennessee, Knoxville, Tennessee, United States of America, **2** Oak Ridge National Laboratory, Oak Ridge, Tennessee, United States of America

* dhayes1@utk.edu (DGH); pingalis@ornl.gov (SVP)

**Data Availability Statement:** All relevant data are available from Dryad with the following DOI: 10.5061/dryad.sf7m0cg3m.

## Abstract

Terrestrial nanoplastics (NPs) pose a serious threat to agricultural food production systems due to the potential harm of soil-born micro- and macroorganisms that promote soil fertility and ability of NPs to adsorb onto and penetrate into vegetables and other crops. Very little is known about the dispersion, fate and transport of NPs in soils. This is because of the challenges of analyzing terrestrial NPs by conventional microscopic techniques due to the low concentrations of NPs and absence of optical transparency in these systems. Herein, we investigate the potential utility of small-angle neutron scattering (SANS) and Ultra SANS (USANS) to probe the agglomeration behavior of NPs prepared from polybutyrate adipate terephthalate, a prominent biodegradable plastic used in agricultural mulching, in the presence of vermiculite, an artificial soil. SANS with the contrast matching technique was used to study the aggregation of NPs co-dispersed with vermiculite in aqueous media. We determined the contrast match point for vermiculite was 66% $D_2O$ / 33% $H_2O$. At this condition, the signal for vermiculite was ~50–100%-fold lower that obtained using neat $H_2O$ or $D_2O$ as solvent. According to SANS and USANS, smaller-sized NPs (50 nm) remained dispersed in water and did not undergo size reduction or self-agglomeration, nor formed agglomerates with vermiculite. Larger-sized NPs (300–1000 nm) formed self-agglomerates and agglomerates with vermiculite, demonstrating their significant adhesion with soil. However, employment of convective transport (simulated by *ex situ* stirring of the slurries prior to SANS and USANS analyses) reduced the self-agglomeration, demonstrating weak NP-NP interactions. Convective transport also led to size reduction of the larger-sized NPs. Therefore, this study demonstrates the potential utility of SANS and USANS with contrast matching technique for investigating behavior of terrestrial NPs in complex soil systems.

**Funding:** DGH, HON and BRE received seed funding from the University of Tennessee Institute for a Secure and Sustainable Environment. AFA and DGH received seed funding from the University of Tennessee Institute of Agriculture and Mobius, Inc (Lenoir City, TN). The funders had no role in study design, data collection and analysis, decision to publish, or preparation of the manuscript.

**Competing interests:** Mobius, PLC (Lenoir City, TN, USA) kindly provided support for this research project by partially funding the salary of Mr. Anton F. Astner, graduate (PhD) student. The research work described in our manuscript does not involve any intellectual property developed by Mobius. The relationship between Anton F. Astner and Douglas G. Hayes does not alter the authors' adherence to all PLOES ONE policies on sharing data and materials.

# Introduction

Increasing global plastic pollution is an emerging threat to marine and terrestrial ecosystems worldwide [1, 2]. In 2015, global plastic production exceeded 400 million metric tons (MMT), from which 300 MMT of plastic waste were formed [3]. Approximately 79% of plastic waste generated has been dispersed into the environment through improper disposal or landfilled [3]. Macro- and mesoplastics (average particle size, or $d_p$ of > 25 mm and 5–25 mm, respectively) undergo further size reduction, resulting in microplastics (MPs) and nanoplastics (NPs), possessing $d_p$ of 1–5000 μm and 1–1000 nm, respectively [1, 4, 5]. A primary environmental concern is that MPs and NPs, due to their hydrophobicity, are likely to carry adsorbed contaminants such as pesticides, plasticizers, or other potentially harmful agents that can directly impact the marine and terrestrial organisms through uptake and digestion [6, 7]. Recent studies reported potential harm to marine biota such as microorganisms [8–12], fish and other organisms [6, 13–16]. MP contamination has been reported in foods, including fish, bivalves, crustaceans and beverages [17]. However, the effects of MPs and NPs on terrestrial environments has been rarely studied [1, 18]. Due to their smaller size than average soil particles, NPs are likely to be mobile and propagate up to higher ranks in food chains via adsorption through plant roots and subsequent uptake by livestock [19]. NPs are sufficiently small to be able to enter organs and cross the brain-blood and placental barriers [17]. However, plastic-soil interactions are not fully understood [11].

A significant secondary source of MPs and NPs in agricultural soils are plastic mulch films, which are used for the production of vegetables and other specialty crops. They serve as a barrier applied to the soil surface to prevent weeds and evaporative loss of water, sustain soil temperature, reduce diseases, and pests [20–22]. The most frequently used plastic mulch material is polyethylene (PE); however, there are no desirable end-of-life. Recycling programs are mostly unavailable [20]. Furthermore, PE fragments commonly form as a result of their embrittlement via environmental weathering, particularly by solar radiation, and persist in the environment for many years since PE is poorly biodegradable. As an alternative to PE mulches, biodegradable plastic mulches (BDMs) are designed to be tilled into soil after the crop harvest, where they are expected to ultimately be decomposed by microorganisms [20–22]. The most prominent polymers used for BDMs are poly(butylene succinate) (PBS), poly(butylene succinate-co-adipate) (PBSA), and poly(butylene-adipate-co-terephthalate) (PBAT), thermoplastic starch, cellulose, polylactic acid and polyhydroxyalkanoates [20, 22].

Although BDMs should be completely catabolized into $CO_2$ and water by soil microorganisms [23, 24], in practice, inconsistencies in their breakdown and biodegradation have been observed [25–27]. Moreover, biodegradation in soil is typically slow: 90% biodegradation in two years via standardized lab testing is a criterion recently issued by the European Union for BDMs [28]. MPs have been detected at levels of 10–20 kg/ha in fields where BDMs were used continuously for vegetable production [29].

Also, MPs formed from BDMs may be a source of terrestrial NPs [30]. NPs They have not been detected in soils to date, mainly due to the absence of a robust analytical approach, although they are likely to occur [11]. NPs derived from BDMs in soil may negatively impact soil health, fertility, and crop production and would be more likely than MPs to enter the food production system due to their small size. NPs are expected to behave differently than MPs due to their anticipated colloidal behavior, e.g., the ability to undergo Brownian motion, and differently than soil micro- and nanoparticles (which particularly occur in clays) due to their more hydrophobic nature than most soils. (But, it is noted that adsorption of hydrophobic molecules onto soils can induce hydrophobicity into soils [31].) In addition, the density of NPs

for agricultural plastics would significantly lower than soils: 0.5–1 g cm$^{-3}$ for plastics used in mulch films versus particle densities of 2–3 g cm$^{-3}$ for many soils [26, 32].

For risk assessments and remediation, information about the impacts, distribution, behavior, and fate of terrestrial NPs is essential. However, their detection is difficult due to their nanoscale dimensions and their relatively small concentration compared to soil particulates. Microscopic detection is possible in soils for fluorescently-derivatized NPs, for instance. However, the introduction of a fluorophore into an NP may change its properties and introduce artifacts into the sample analysis. In this paper, we introduce analysis by small-angle neutron scattering (SANS) and ultra-SANS (USANS) as a potentially valuable approach to measure the aggregation behavior of NPs in solution and in the presence of soil particulates. Unlike microscopy, SANS allows for *in situ* measurements of size, shape, and agglomeration of NPs and soil, and neutrons are non-destructive to samples. Another advantage offered by SANS and USANS methods is the use of neutron contrast matching to isolate the behavior of one nanoscale component from that of the other components in the neutron beam.

Specifically, this investigation tests the proof-of-concept that SANS and USANS can be used in conjunction with neutron contrast matching to isolate the signal of NPs from that attributable to soil. The contrast match point (CMP) of vermiculite microparticles, an artificial soil similar in particle size to silt [33], was determined via SANS analysis of aqueous suspensions at various $H_2O/D_2O$ ratios. The CMP refers to the level of deuteration in solvent (water) that minimizes vermiculite's signal. Then, suspensions of NPs formed from PBAT-based BDMs and vermiculite in water at the CMP were examined to investigate their agglomeration behavior *in situ*. The effect of *ex situ* stirring before SANS and USANS analysis was investigated to determine the effect of convective transport on agglomeration. The SANS and USANS measurements test the hypothesis that the NPs are more likely to agglomerate with soil than to self-agglomerate. The agglomeration behavior may play a key role in the long-term fate, transport and biodegradability of terrestrial NPs, especially at the water-soil interface. Particle agglomeration of NPs would also impact NPs' migration in surface waters and may explain the inability to detect NPs by flotation or leaching of soil samples.

## Materials and methods

### Materials

BioAgri, a black-colored BDM film prepared from Mater-Bi® (Grade EF04P), a starch-copolyester blend containing PBAT as its principal constituent, was kindly provided by BioBag Americas (Dunevin, FL, USA). The film referred to as "PBAT" in this paper, possesses an apparent density of 22.81±0.411 g m$^{-2}$, a thickness of 29±1.2 μm (i.e., a specific gravity of 0.787), a peak load of 12.05± 0.586 N, an elongation of 295±30% at maximum tensile stress in the machine direction and a contact angle of 82.5±1.1 [26]. Other physicochemical properties are given in the cited reference. The original film was provided as a 1.22 m-wide roll and stored at 20.6 ± 2.1˚C and 61.8 ± 10.6% relative humidity. Deuterium oxide ($D_2O$) was purchased from Acros (Geel, Belgium). Deionized water was used throughout all experiments. Vermiculite ($Mg_{1.8}Fe^{2+}_{0.9}Al_{4.3}SiO_{10}(OH)_2$*$4H_2O$), Grade 4, mesh size 7.9 mm, was purchased from Uline (Pleasant Prairie, WI, USA). Raw vermiculite particles possessed an average particle size of 4.65 ± 2.39 mm (L/W ratio 1.39, measured with ImageJ software [34]) and were comminuted with a pestle grinder and sieved through a cascade of four sieves (W.S. Tyler, Cleveland, OH, USA) with mesh sizes of #20 (840 μm), #60 (250 μm), #140 (106 μm), and #325 (45 μm). The 45 μm sieving particle fraction was collected, and an average particle size of 38±12 μm was measured using a model SZ 61 stereomicroscope from Olympus (Shinjuku, Tokyo, Japan) with a Digital Sight DS-Fi1 integrated with a camera head from Nikon (Shinagawa, Tokyo,

Japan). Soil particles of this size were selected because of their effective dispersion in water, and their high monodispersity was anticipated to simplify interpretation of the SANS data. Vermiculite particles within the given size range mimic silt [33]. Image analysis was performed using ImageJ software [34] by converting micrographs into 8-bit images (representing 28 gray levels) using a proper threshold setting (dividing the image into two or more classes of pixels). The subsequent analysis included the binary file conversion of the adjusted image. The average diameter, $d_p$, was estimated using the Image J's "analyze particles" algorithm. A representative image of the entire sample was collected and processed though Image J using one replicate. For each particle size fraction, 250 particles were counted and averaged.

## Production of NPs

NPs were prepared from PBAT film according to the optimized procedure [30]. PBAT specimen (~1.0 g), cut from BDMs films to dimensions of ~120 mm (machine direction) x 20 mm (cross direction), were presoaked in water (800 mL) for 5 min, recovered and transferred to a cryogenic container filled with liquid nitrogen (200 mL) and soaked for 5 min. The frozen PBAT film fragments (1.0 g) were transferred into an Osterizer type blender (Oster Accurate Blend 200, Boca Raton, FL, USA), and dry-comminuted for 10 s. Water (400 mL) was added to the PBAT fragments to form a slurry, and then the blender was operated for 10 min at $10 \times 10^{-3}$ $min^{-1}$. After blending, the slurries were filtered under vacuum through a paper membrane filter (1 μm mesh) using a Büchner funnel apparatus and then air-dried for 48 h to reduce moisture to < 1%. The resulting MP fragments were possessed $d_p$ of 1.47 ± 0.45 mm (ImageJ analysis of stereomicrographs) [30]. The cryogenically embrittled PBAT MPs were fed to a rotary mill (Model 3383-L10 Wiley Mini Mill, fitted with screen, Arthur H. Thomas Co., Philadelphia, PA, USA) by using sieve sizes of 20 mesh (840 μm) for the first pass and 60 mesh (250 μm) for the second pass through the mill. The residence time for milling was 20 min per pass. MPs recovered from milling were fractionated via a cascade of four sieves (W.S. Tyler, Cleveland, OH, USA) with mesh sizes of #20 (840 μm), #60 (250 μm), #140 (106 μm), and #325 (45 μm). Uniform particle size distributions were achieved by mounting the sieves on an Eppendorf thermomixer 5350 (Hamburg, Germany) and shaking for 30 min at 300 rpm.

The 106 mm sieve fraction was suspended in an aqueous slurry (4.0 L) via magnetic stirring at 400 rpm for 24 h, thereby providing a 1% solution of MPs. After stirring, slurries were subjected to the wet-grinding process using a "supermass colloider" (MKCA6-2, Masuko Sangyo, Tokyo, Japan) at a speed of 1500 rpm and 27 subsequent passes (collection of particles and reintroduction into the colloider) to provide a uniform particle size reduction. The slurry recovered from wet-grinding was transferred to a 1000 mL beaker and magnetically stirred for 4 h (300 rpm at 25ºC). The final concentration of the slurry aliquot was 0.37 (wt)%. The resultant particles were vacuum dried at 40˚C for 48 h and stored in an air-sealed container. The dried NPs possessed an average $d_p$ of 366.0±0.65 nm according to dynamic light scattering (DLS) analysis (bimodal distribution: $d_p$ values of 536.8±151.8 nm and 63.8±13.7 nm, with each subpopulation's distribution described by a two-parameter Weibull distribution) [30], and were used for SANS/USANS sample preparation. The NPs' zeta potential (in $H_2O$) was determined to be -22±3.6 mV through employment of a Zetasizer Nano instrument (Malvern Instruments, Malvern, UK) using a Smoluchowski model. According to Atomic Force Microscopic (AFM) analysis, performed using a model Multimode 8 instrument from Bruker (Santa Barbara, CA, USA), NPs were irregularly shaped and possessed an average roughness of 12.22 ±1.55 nm (S1 Fig). The pH (electrical conductivity) value for the 0.5% vermiculite slurry in water was determined to be 10.14±0.02 (89.57±0.28 μS $cm^{-1}$) and after the addition of 1% NP to the 0.5% vermiculite slurry to be 9.54±0.13 (80.03±0.29 μS $cm^{-1}$).

## Sample preparation for SANS and USANS experiments

SANS samples consisted of slurries containing 1.0 wt% PBAT NPs and/or 0.5% vermiculite in different ratios of $H_2O/D_2O$ solvent. *Ex situ* stirring was employed for several samples by mixing slurries (1.0 mL) in 7 mL borosilicate glass scintillation vials at 400 rpm (radius = 1.5 cm) for 24 h at 20 ± 1˚C using a 4-sample stirrer (Isotemp 60 Fisher Scientific, Pittsburgh, PA, USA). Upon completion of stirring, samples were recovered and kept refrigerated prior to SANS/USANS analysis. Changes in $d_p$ due to refrigeration were within 5% (DLS analysis).

## SANS and USANS analysis

SANS and USANS experiments were conducted at 22 ± 1˚C using the Bio-SANS and USANS instruments at Oak Ridge National Laboratory (ORNL), Oak Ridge, TN USA. Further details on the instrumentation and their settings are provided elsewhere [35–37]. The NP/vermiculite slurries were loaded into 1.0 mm pathlength titanium cells. To obtain an even distribution of NPs and vermiculite, the cells were gently rotated *in situ* in the radial direction using a tumbling sample changer at 10 rpm and 5 rpm for SANS and USANS, respectively, to enable uniform dispersion in the path of the neutron beam [37]. Moreover, the tumbling speed was set to match the settling velocity of particles to ensure that the particles remain mostly in the path of the beam, rather than settle out. The incident wavelengths were at 6.09 Å for SANS and 3.6 Å for the primary USANS beam. The higher order neutron energies from the Bragg reflections for USANS were separated from the primary beam (3.6 Å) by time-of-flight, allowing for the elimination of a major source of background in this class of instrument. The scattering from these samples was not sufficiently strong for the data to benefit from the additional information potentially provided by scattering from these reflections at extremely low *Q*. SANS experiments employed a single configuration with the main detector at 15.5 m and the wing detector at 1.4˚ rotation to allow for an effective range for the momentum transfer, $Q$ (= $4\pi\lambda^{-1}\sin[\theta/2]$, where $\theta$ is the scattering angle and $\lambda$ is the wavelength of incident neutrons, 6.09 Å), of 0.003–0.50 Å$^{-1}$. USANS employed a 30 m detector distance to produce a *Q* range of 5 x 10$^{-5}$–2 x 10$^{-3}$ Å$^{-1}$. Typical acquisition times were 0.5–1.0 h and 8–12 h for SANS and USANS, respectively. We did not observe the settling out of particles at any instance during the SANS or USANS experiments. Although we cannot fully rule out that particle aggregation was induced by low *in situ* tumbling, the absence of settling gives us confidence that the impact of this event was small. Error bars given in the figures for *I(Q)* are based on counting statistics. The square root of the counts and subsequently, error propagation, were applied for any downstream corrections to the data.

SANS data (scattered intensity *I(Q)* vs. *Q*) were reduced using Mantid software and analyzed by fitting the data with a nonlinear general scattering law based on form and structure factors [*P(Q)* and *S(Q)*, respectively] through an Igor Pro macro written by NIST staff scientists [38]. USANS data was de-smeared using a slit height of 0.042 Å$^{-1}$ (in units of momentum transfer) using NIST USANS package (Igor Pro) prior to merging SANS and USANS data. The merge process was performed via determination of the best power law line that fit both sets of data [36, 38]. A power law fit was applied to the linear portions of the combined SANS and USANS data (Porod region, $1/2\,Q\,d_p >> 1$).

$$I(Q) = \alpha Q^{-\beta} \tag{1}$$

For $0 \leq \beta \leq 3$, $\beta$ is the power-law exponent and represents the mass fractal dimension ($D_f$). When the power-law exponent varies as $3 \leq \beta \leq 4$, then surface fractal ($D_s$) varies as $3 \geq D_s \geq 2$ ($D_s = 6-\beta$). $\beta = 3$ (or $D_s = 3$) represents a rough surface, while $\beta = 4$ (or $D_s = 2$) represents a smooth surface [39].

After subtraction of Eq 1, the resultant "excess" data $(I(Q)-\alpha Q^{-\beta})$ was fitted using form factor-structure factor modeling [40].

The structure factor, simulating particle-particle interactions, was assumed to be 1.0 due to the small volume fraction of NPs. A polydisperse sphere form factor was employed, providing the average particle radius, $R$, the polydispersity of the radius (based on a Schulz distribution) $pd$, and the volume fraction of dispersed phase $\phi$ as outputs and the scattering length densities of PBAT and water at different levels of deuteration as inputs [41]. $B_{incoher}$ was set equal to 0.0 since incoherent contribution was subtracted during reduction of data. The average particle diameter of the NPs, $d_p$, is therefore equal to 2R.

## Results and discussion

### Determination of the contrast match point (CMP) for vermiculite

Slurries of vermiculite (0.5%) in water consisting of various proportions of $D_2O$ were analyzed by SANS to determine the contrast match point. As shown in Fig 1, the scattered intensity, $I(Q)$, decreased as the $D_2O$ fraction was increased up to 60% v/v; then, further increases in $D_2O$ concentration increased $I(Q)$. The data reflects a power law relationship (per Eq 1), with $\beta$ decreasing from 3.4 to 2.9 as the $D_2O$ content was decreased from 100% to 60% and increased from 0% to 60%, approaching a minimum at 60% $D_2O$ (Fig 1A). The values of $\beta$ are comparable to the values reported for small-angle x-ray scattering analysis of vermiculite [42] and represent a rough surface.

We determined the CMP for vermiculite by plotting the square root of $I(Q)$ in the low-$Q$ region (0.004 Å$^{-1}$) vs. volume fraction of $D_2O$ in the solvent. According to this plot (Fig 1B), the CMP is ~67% $D_2O$/33% $H_2O$, corresponding to a neutron contrast of 4.08 x 10$^{-6}$ Å$^{-2}$. Fig 1A contains the SANS data at the CMP. Although $I(Q)$ for vermiculite is decreased nearly 100-fold at the CMP relative to 100% $H_2O$ and over 10-fold compared to 100% $D_2O$, the signal is not entirely removed. The inability to completely suppress the scattering is likely a result of the spatial heterogeneity of vermiculite's scattering length density, due to heterogeneity in the particle density and chemical composition.

### Effect of *ex situ* stirring and vermiculite on NP structure and agglomeration

SANS and USANS analyses at the CMP determined the impact of *ex situ* stirring (24 h) on the agglomeration of NPs in the presence of vermiculite. The addition of vermiculite led to a slight decrease of $I(Q)$, confirming that contrast matching minimized the scattering attributable to vermiculite and that NPs were removed from the neutron beam through agglomeration with vermiculite (Fig 2A, inset). The power-law exponent $\beta$ (Eq 1) did not change appreciably with stirring: 3.5±0.1, a value that suggests the surface characteristics of the NPs are rough (Fig 2B). *Ex situ* stirring increased the intensity of the SANS signal of PBAT NPs, a result suggesting that convection improved the dispersion of the NPs by disrupting the formation of large agglomerates. An alternate explanation would be that convection increased the extent of solvent penetration into NPs and their agglomerates. The addition of vermiculite reduced the extent of the increase for $I(Q)$.

The subtraction of the power law relationship (Eq 1) from $I(Q)$, referred to herein as "excess" scattering, reveals the presence of scattering intensity oscillations of NPs and their agglomerates for both USANS and SANS data (Fig 3A and 3B). The "excess" oscillations were fitted with spherical form factor models (Schultz distribution to account for polydispersity in the radius) as a first approximation. Values of the volume fraction of dispersed phase (i.e., of

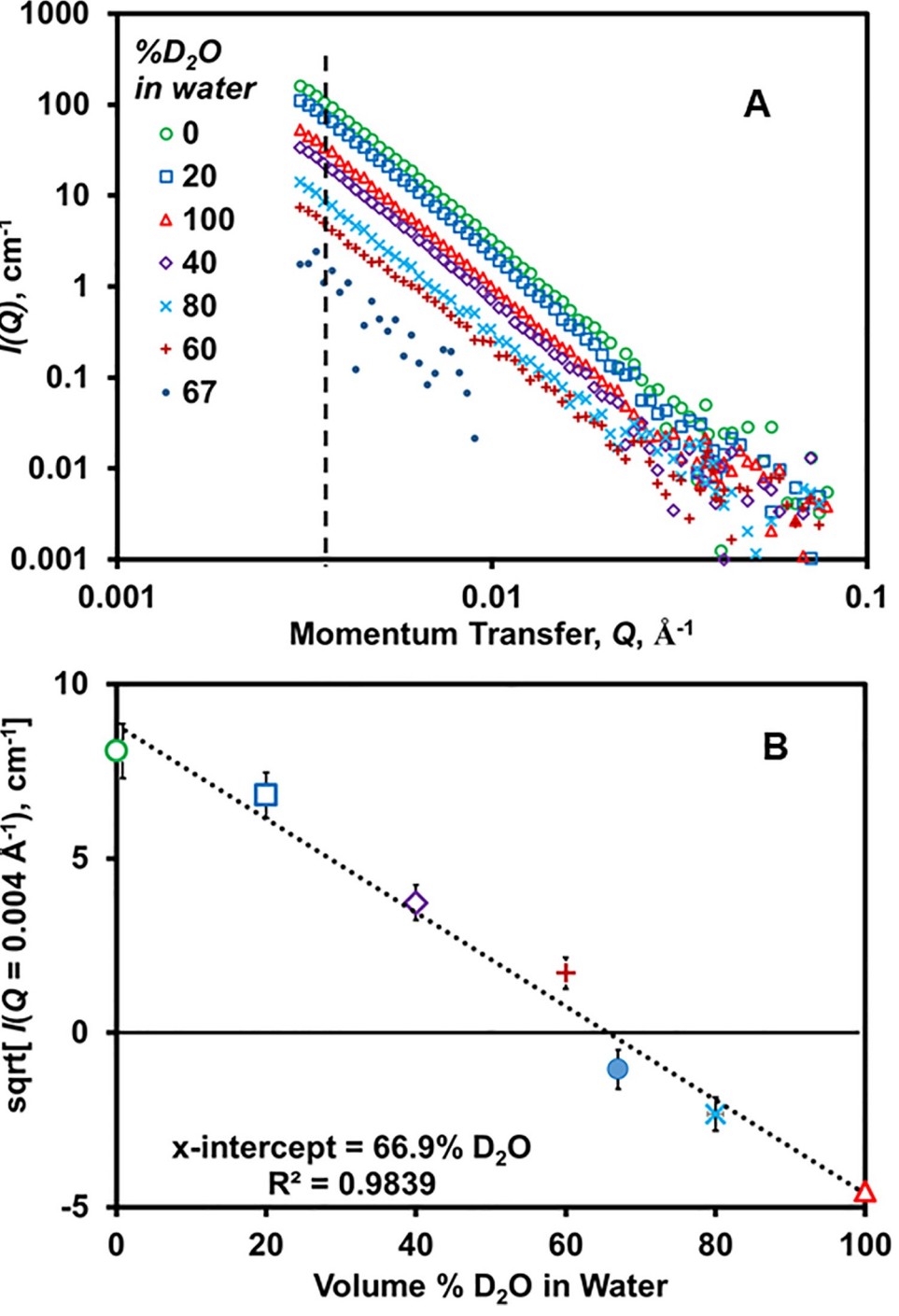

**Fig 1. Determination of the neutron contrast match point for vermiculite (0.5 wt % dispersed in $H_2O/D_2O$ mixtures).** (A) $I(Q)$ vs $Q$ data, (B) square root of $I(Q)$ at $Q = 0.004$ Å$^{-1}$ vs $D_2O$ volume % in water. Error bars for Fig A are provided in S2 Fig.

NPs; φ), $d_p$, and polydispersity ($pd$) are given in Table 1. The "excess" curves at high-$Q$ from SANS likely correspond to individual NPs, with $d_p$ being ~51.7 nm (Table 1). According to DLS analysis, NPs (in the absence of vermiculite) possessed a bimodal size distribution, with the smaller size subpopulation possessing $d_p$ of ~50–65 nm, comparable to the SANS-derived

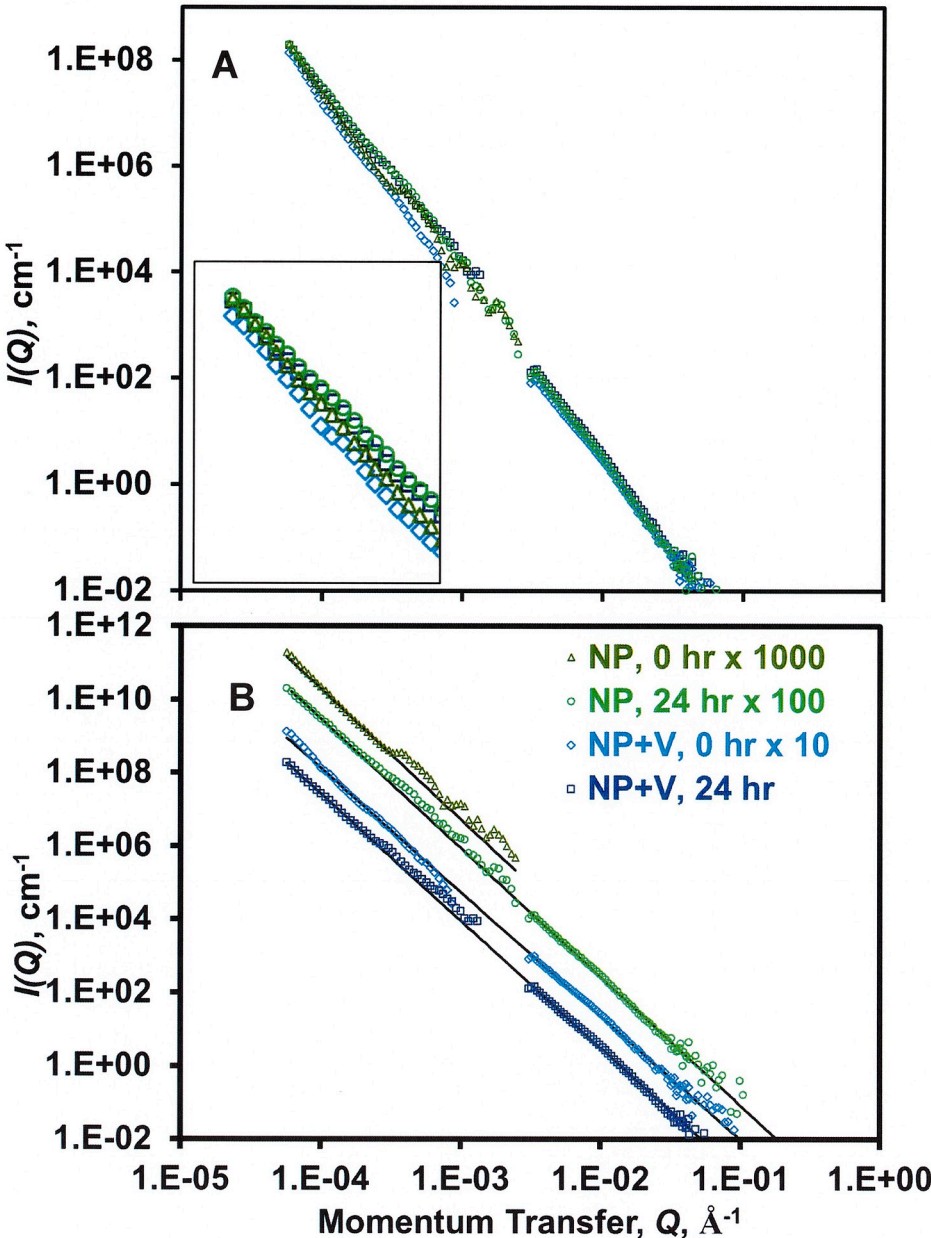

**Fig 2. Effect of *ex situ* stirring and the presence vs. absence of vermiculite (*V*) on SANS data for NPs at the contrast match point for vermiculite (67% D$_2$O in water; cf. Fig 1).** NPs and *V* were present in the suspension at 1.0 wt% and 0.5 wt%, respectively. **(A)** SANS and USANS data (inset: an expansion of the data at low *Q*) and **(B)** power law fitting (Eq 1) of data in Fig A [$I(Q) = \alpha Q^{-\beta}$, where $\beta$ = 3.4–3.6]. For Fig B, $I(Q)$ was multiplied by a constant (as given in the legend) to improve visualization. Error bars are smaller than the size of the symbols.

value [30]. The absence of variance for the "excess" oscillations with *ex situ* stirring and the addition of vermiculite suggests the smaller-sized subpopulation of NPs are well dispersed in water and are unlikely to form agglomerates (Fig 3A and Table 1).

The "excess" USANS data reflects the presence of dispersions of $d_p > 300$ nm (Fig 3A and Table 1), which likely correspond to the larger, $d_p = 537$ nm, sub-population of the bimodal distribution observed by dynamic light scattering [30]. A shoulder at low-$Q$ (0.5–1.0 x 10$^{-4}$ Å$^{-1}$) is

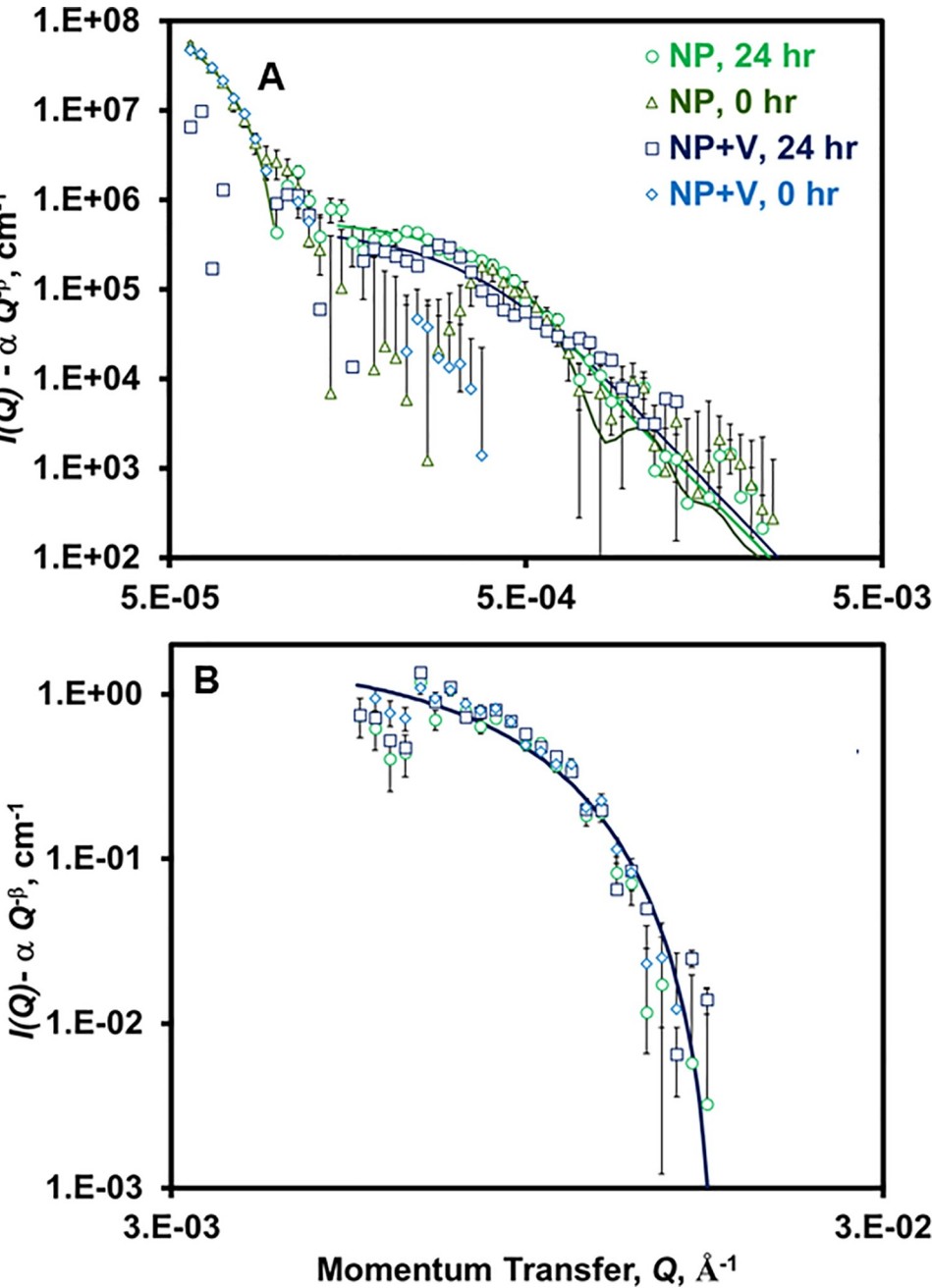

**Fig 3.** Schulz polydisperse sphere form factor model fitting of (A) USANS and (B) SANS "excess" scattering data ($I(Q)$ from Fig 2A minus power law fit from Fig 2B). Model parameters are given in Table 1. V and NP refers to vermiculite and nanoplastics, respectively. Experimental conditions are given in Fig 2.

believed to represent NP self-agglomerates of the larger sized NP subpopulation (8.5 μm = $d_p$). Moreover, vermiculite particles possess a $d_p$ of ~38 μm, which would produce a USANS signal for its agglomerates at a lower $Q$ value than available through the USANS instrument. *Ex situ* stirring diminished the low-$Q$ "excess" shoulder, a result suggesting that convection will break apart the NP-NP attractions that lead to agglomeration (Fig 3A).

The oscillations within the $Q$ range of $0.2$–$1.4 \times 10^{-3}$ Å$^{-1}$, likely reflecting NPs of the larger subpopulation, undergo a decrease in size and an increase of *pd* upon *ex situ* stirring (Fig 3A

**Table 1. Results from model fitting of SANS+USANS "excess" data plotted in Fig 3[a,b].**

| Composition [c] | Q range, Å$^{-1}$ | $\phi$ x 10$^4$ | $d_p$, nm | pd |
|---|---|---|---|---|
| NP, 0 hr | 0.5–1.0x10$^{-4}$ | 9.7±1.2 | 8710±20 | 0.01±0.01 |
| NP+V, 0 hr | 0.5–1.0x10$^{-4}$ | 9.3±1.2 | 8330±20 | 0.08±0.01 |
| NP, 0 hr | 0.4–2.4x10$^{-3}$ | 12.0±1.0 | 1060±40 | 0.11±0.03 |
| NP, 24 hr | 0.2–1.4x10$^{-3}$ | 14.0±1.0 | 792±40 | 0.32±0.03 |
| NP+V, 24 hr | 0.2–1.4x10$^{-3}$ | 7.98±1.0 | 287±30 | 0.84±0.05 |
| NP, 24 hr | 0.5–1.7x10$^{-2}$ | 0.30±0.05 | 51.0±3.0 | 0.00±0.01 |
| NP+V, 0 hr | 0.5–1.7x10$^{-2}$ | 0.39±0.05 | 51.0±3.0 | 0.00±0.01 |
| NP+V, 24 hr | 0.5–1.7x10$^{-2}$ | 0.40±0.05 | 51.0±3.0 | 0.00±0.01 |

[a] Obtained from fitting "excess" data with a form factor based on polydisperse spheres with radii possessing a Schulz distribution. The structure factor was assumed to be ≈1.0. The scattering length density of the spheres (vermiculite) and solvent (D$_2$O/H$_2$O 67/33 v/v) were held constant at 1.60 and 4.08 x 10$^{-6}$ Å$^{-2}$; the incoherent background was assumed to equal zero because all incoherent background was subtracted during data reduction

[b] column headings: $\phi$ and $d_p$ are the volume fraction and average diameter of dispersed NPs, respectively; pd = polydispersity index (for radii); [c] NPs and V represent PBAT NPs (1 wt%) and vermiculite (0.5 wt%), respectively.

and Table 1), suggesting the NPs undergo size reduction. It is unclear if this trend may be attributable to the breakup of agglomerates composed of smaller-sized NPs by convection. The slight increase of $\phi$ for the smaller-sized ($d_p$ = 51 nm) NP subpopulation with *ex situ* stirring (Table 1) suggests the latter event may occur to a small extent. Although NPs of this subpopulation decreased in size, the USANS scattering peak increased in the low-$Q$ range (0.2–0.4 x10$^{-3}$ Å$^{-1}$), suggesting that *ex situ* stirring increased the effective dispersion of larger-sized NPs. The increase of $\phi$ (Table 1) for the main USANS oscillation peaks supports this hypothesis. The USANS "excess" oscillations for NPs in the presence of vermiculite after *ex situ* stirring are lower than in the latter's absence (e.g., $\phi$ decreased 2-fold), suggesting that NPs of the larger subpopulation form agglomerates with vermiculite (particularly larger NPs thereof, noted by the decrease of $d_p$) (Fig 3A and Table 1). Therefore, the data demonstrate the direct interaction between NPs and soil particulates, the extent of which is modulated by convection and increases for large-sized NPs.

## Conclusions

This paper describes a preliminary study to demonstrate the potential utility to employ SANS and USANS with neutron contrast matching to investigate the behavior of NPs in terrestrial systems. We determined the CMP of vermiculite, an artificial soil, providing conditions where the scattering contribution of vermiculite would be minimized and investigated the impact of soil and convective transport on NPs derived from a biodegradable plastic mulch film, composed of PBAT. Results suggest that NPs of larger size self-associate and also aggregate with soil, with convection minimizing the agglomeration. The larger-sized NPs (which may partially consist of NP aggregates) undergo size reduction under convection, while smaller-sized NPs (51 nm) remained intact.

NPs are an emerging threat to soil, particularly agricultural soils, due to their involvement with producing the world's food supply and the prominence of plastic in vegetable and fruit production systems, particularly as mulch film. Their hydrophobicity is known to drive NPs' adsorption of toxicants such as pesticides and phthalate-based plasticizers, which can enter food supplies. Even biodegradable plastics, known to form MPs, will likely form NPs that will reside in the soil for at least several months. There exists a critical gap in fundamental

understanding of terrestrial NPs and their potential impact on soil fertility, terrestrial organisms such as earthworms, and microbial communities, as well as their long-term fate and transport (including to groundwater). Such information is necessary to design strategies for mitigation. NPs are challenging to investigate in soils due to their low concentration and the solid-phase nature of the system. SANS and USANS, with contrast matching, may serve as a robust approach, that will allow for direct measurements of size and agglomeration behavior of NPs under environmentally relevant conditions. We are currently evaluating the effect of NP concentration and *ex situ* stirring time on the agglomeration of NPs and soil by SANS and USANS using the approach described herein.

## Supporting information

**S1 Fig. AFM images used for ImageJ measurement of surface roughness for nanoplastics.**
(PDF)

**S2 Fig. Data of Fig 1A of the main paper replotted to include error bars.**
(PDF)

## Acknowledgments

BioBag Americas, Inc. (Dunevin, FL, USA), kindly provided the PBAT-based biodegradable mulch film employed for preparing NPs. We thank Dr. Li Teng (Department of Food Science and Technology, University of Tennessee) for his technical assistance in performing AFM analysis of NPs. Neutron scattering research conducted at the Bio-SANS instrument, a DOE Office of Science, Office of Biological and Environmental Research Structural Biology Resource, and USANS (BL-1A), a DOE office of Science, office of Basic Energy Sciences resource, used resources at the High Flux Isotope Reactor and the Spallation Neutron Source, respectively, a DOE Office of Science, Scientific User Facilities operated by the Oak Ridge National Laboratory (ORNL).

ORNL is managed by UT-Battelle, LLC, for the U. S. Department of Energy under Contract DE-AC05-00OR22725. This manuscript has been authored by UT-Battelle, LLC, under Contract No. DE-AC05-00OR22725 with the U.S. Department of Energy. The United States Government retains and the publisher, by accepting the article for publication, acknowledges that the United States Government retains a non-exclusive, paid-up, irrevocable, world-wide license to publish or reproduce the published form of this manuscript, or allow others to do so, for United States Government purposes. The Department of Energy will provide public access to these results of federally sponsored research in accordance with the DOE Public Access Plan (http://energy.gov/downloads/doe-public-access-plan).

## Author Contributions

**Conceptualization:** Anton F. Astner, Douglas G. Hayes, Sai Venkatesh Pingali, Hugh M. O'Neill, Barbara R. Evans, Volker S. Urban.

**Data curation:** Anton F. Astner, Douglas G. Hayes, Sai Venkatesh Pingali.

**Formal analysis:** Anton F. Astner, Douglas G. Hayes, Sai Venkatesh Pingali.

**Funding acquisition:** Douglas G. Hayes, Hugh M. O'Neill, Barbara R. Evans.

**Investigation:** Anton F. Astner, Douglas G. Hayes, Sai Venkatesh Pingali, Kenneth C. Littrell.

**Methodology:** Anton F. Astner, Douglas G. Hayes, Sai Venkatesh Pingali, Hugh M. O'Neill, Kenneth C. Littrell, Volker S. Urban.

**Project administration:** Douglas G. Hayes, Sai Venkatesh Pingali.

**Resources:** Douglas G. Hayes, Sai Venkatesh Pingali, Hugh M. O'Neill, Kenneth C. Littrell, Volker S. Urban.

**Software:** Douglas G. Hayes.

**Supervision:** Douglas G. Hayes.

**Validation:** Douglas G. Hayes, Sai Venkatesh Pingali.

**Visualization:** Anton F. Astner, Douglas G. Hayes, Sai Venkatesh Pingali.

**Writing – original draft:** Anton F. Astner, Douglas G. Hayes, Sai Venkatesh Pingali, Hugh M. O'Neill.

**Writing – review & editing:** Anton F. Astner, Douglas G. Hayes, Sai Venkatesh Pingali, Hugh M. O'Neill, Barbara R. Evans.

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
