## [Decision Letter · Decision Letter 0]

3 Feb 2020

PONE-D-19-33737

Effects of Soil Particulates and Convective Transport on Dispersion and Aggregation of Nanoplastics via Small-Angle Neutron Scattering (SANS) and Ultra SANS (USANS)

PLOS ONE

Dear Dr Professor Hayes,

Thank you for submitting your manuscript to PLOS ONE. After careful consideration, we feel that it has merit but does not fully meet PLOS ONE’s publication criteria as it currently stands. Therefore, we invite you to submit a revised version of the manuscript that addresses the points raised during the review process.

We would appreciate receiving your revised manuscript by Mar 19 2020 11:59PM. To enhance the reproducibility of your results, we recommend that if applicable you deposit your laboratory protocols in protocols.io, where a protocol can be assigned its own identifier (DOI) such that it can be cited independently in the future. For instructions see: http://journals.plos.org/plosone/s/submission-guidelines#loc-laboratory-protocols

We look forward to receiving your revised manuscript.

Kind regards,

Pratheep K. Annamalai

Academic Editor

PLOS ONE

Journal Requirements:

"DGH, HON and BRE received seed funding from the University of Tennessee Institute for a Secure and Sustainable Environment. AFA and DGH received seed funding from the University of Tennessee Institute of Agriculture and Mobius, Inc (Lenoir City, TN). The funders had no role in study design, data collection and analysis, decision to publish, or preparation of the manuscript."

We note that you received funding from a commercial source: Mobius, Inc.

4. Please include a copy of Table 3 which you refer to in your text on page 13.

Reviewers' comments:

Reviewer's Responses to Questions

**Comments to the Author**

1. Is the manuscript technically sound, and do the data support the conclusions?

Reviewer #1: Partly

Reviewer #2: Yes

2. Has the statistical analysis been performed appropriately and rigorously? 

Reviewer #1: Yes

Reviewer #2: No

3. Have the authors made all data underlying the findings in their manuscript fully available?

Reviewer #1: Yes

Reviewer #2: Yes

4. Is the manuscript presented in an intelligible fashion and written in standard English?

Reviewer #1: Yes

Reviewer #2: Yes

5. Review Comments to the Author

Reviewer #1: General comments

I thank the authors for this highly interesting and provocative work. I hope that my review, although quite negative, does not discourage these investigations. The work describes small neutron angle scattering over an extended q-range from samples that are to act as proxies for the convective dispersive transport of nano-plastics in soil: vermiculite and a commonly biodegradable plastic mulch. While the sample preparation, and the neutron scattering experiments, their execution and analysis are highly competent, I believe there is a fundamental misconception in applying the findings of this study to the significant environmental transport of pollutants. This is to say the motivation for this study, as put forward in the document’s text is highly questionable. While it has been well described that many environmental pollutants bind to plastic residues and wastes, this property is not unique to these materials, similar binding properties are well known for that ubiquitous colloidal component of soils, humic materials. There is, as far as I understand, not yet, any way to distinguish the transport properties of “manmade” or “natural” soil carbon. This is to ask the question, “what is it that makes binding to nanoplastics different in ecological importance?” I think the way forward in publishing this elegant experimental study is to place a more general emphasis on this preliminary study in an interesting perspective on the transport of soil carbon using contrast variation to selectively visualise this soil component.

Specific comments

I think personally that level of detail in the fitting procedure obscures from the simple messages of this work.

Page 18 line7

Minor typo “NPs were removed from the neutron be through”, neutron beam?

Page 18 same paragraph

Ambiguous language

“These results suggest that convection improved dispersion (increased surface area) of the

NPs exposed to the neutron beam (e.g., via decreasing the agglomeration of NPs) or increased solvent”

Is there some interaction between with the neutron beam which increases dispersion of particles or is something else meant?

Reviewer #2: Title: Effect of Soil Particulates and Convective Transport on…

Manuscript ID: PONE-D-19-33737

Authors: Astner et al.

Dear Authors,

Thank you for giving me the opportunity to read your article on nanoparticle dispersion. I found the content has an interesting potential. On the other hand, much more clarification and discussion are required to better understand the content and properly evaluate the true value of the work prior to publication. Especially, Materials and Methods should be further clarified. I suggest this article be completely revised before re-submission for the other review processes. As a conclusion, I recommend its major revision at this state.

I hope my comments are helpful.

Good luck,

A reviewer

Major concerns:

1. Materials and Methods

-“apparent density of …g m-2”-> g m-3 ?

-“The 45 μm particle fraction was collected, and an average particle size of 38±12 μm…”->Please provide the justification why this size is appropriate for your study.

-How many particles were counted by ImageJ’s “analyze particles” algorithm?

-“…1.0wt.% PBAT NPs and/or 0.5% vermiculite…”->How did you select these concentrations? Have you also varies them to investigate their effects? According to your introduction, you wish to mimic low concentration of NPs in soil. It seems the particle concentrations are too low, especially for vermiculite to simulate soil. Your samples are like small number of particles swimming in a plenty of water. In agricultural soil, particle movements/displacement should be limited, with much higher volume of soil particles than the one of water.

-Do you have any more information about the particle properties (e.g. degree of hydrophobicity, particle roughness)? What is your sample environment (e.g. solution pH and salt type/concentration)? Nanoparticle dispersions can be strongly affected by those factors.

-How do you justify your ex situ stirring conditions were appropriate/optimized for your study? Stirring conditions should also affect the particle dispersion/aggregation as well as particle concentrations (see my comments above).

-“…samples were recovered and kept for refrigerated prior to SANS/USANS analysis”->Do you have any proof that this procedure can keep the sample status unchanged? Did you feed your refrigerated sample to a SANS/USANS cell after the sample temperature adjusted to your measurement temperature of 22 C?

-“Typical acquisition times were…8-12 h for…USANS…”->Do you have any proof your sample status does not change during such a long period time? Especially, MP samples might sediment during the measurement.

2. Results and discussion

-Figures 1-3: Please add error bars if possible.

-Figure 2-A: Please provide the figure legends.

-Figure 2-B: What does “x10, x100, x1000” mean?

-“Ex situ stirring increased the intensity of the SANS signal of PBAT NPs…These results suggest that convection improved dispersion of the NPs exposed to the neutron beam…”->Does it mean without stirring particles are aggregated and sedimented? If yes, the sample in the beam without experiencing the stirring is not representative to compare with the sample experienced the stirring. If no, why you did not see the change in the scattering curves? Not only the particle numbers, but also particle/aggregate size should also affect the scattering intensity. Referring to your eq.2, the scattering intensity is proportional to (particle volume)2 while (particle number)1.

-“…the attractive forces between NPs must be weak.”->This statement is vague and do not carry any quantitative information. Consider removing it. If you wish to keep it, please provide your proof. See my comments about the particle properties and sample environments.

3. Conclusions

-“NPs are an emerging threat to soil,…as well as their long-term fate and transport (including to groundwater).”->It sounds like an introduction content. Consider moving and integrating it as a part of introduction.

-You may inform possible future work.

Minor concerns:

-Please add line numbers to your manuscript. It makes a reviewer easier to point out his/her concerns.

-Title: “…particulates…”->Consider replacing it with “particles” that is used more in the main text. Please be consistent.

-Introduction: “…plastic plant-soil interactions…”->Do you mean “plastic-soil interactions”?

-“…recycling and landfilling…is impracticable due to soil contamination.”->Not sure what you mean. Can you consider rephrasing it or explaining more?

-Figure 3: Define “Schulz polydisperse form factor” in the main text.

6. PLOS authors have the option to publish the peer review history of their article (what does this mean?). If published, this will include your full peer review and any attached files.

Reviewer #1: Yes: Christopher J. Garvey

Reviewer #2: No

---

## [Author Response · Author response to Decision Letter 0]

19 Mar 2020

19 March 2020

Responses to Editor and Reviewers

On behalf of my coauthors, I thank the reviewers for their kind assistance in providing us comments and suggestions to improve the manuscript. Our responses are given below.

Sincerely,

Douglas G. Hayes, corresponding author.

I. Responses to Editor (Pratheep K. Annamalai, Academic Editor, PLOS ONE)

1. We have prepared our revised manuscript according to the PLOS ONE's style requirements, including those for file naming. 

2. We have revised our Competing Interest Statement to address the involvement of Mobius, Inc. This is provided in our cover letter.

3. We have not made any changes to the Data Availability Statement. We will upload our data as csv files onto Dryad and provide doi numbers for them to PLOS ONE.

4. “Table 3” was a typographical error. “Table 1” should have been used. We have made this correction. Our apologies.

5. We approve of our reviews and responses to the reviewers to be published.

II. Responses to Reviewer 1

Reviewer: I thank the authors for this highly interesting and provocative work. I hope that my review, although quite negative, does not discourage these investigations. The work describes small neutron angle scattering over an extended q-range from samples that are to act as proxies for the convective dispersive transport of nano-plastics in soil: vermiculite and a commonly biodegradable plastic mulch. While the sample preparation, and the neutron scattering experiments, their execution and analysis are highly competent, I believe there is a fundamental misconception in applying the findings of this study to the significant environmental transport of pollutants. This is to say the motivation for this study, as put forward in the document’s text is highly questionable. While it has been well described that many environmental pollutants bind to plastic residues and wastes, this property is not unique to these materials, similar binding properties are well known for that ubiquitous colloidal component of soils, humic materials. There is, as far as I understand, not yet, any way to distinguish the transport properties of “manmade” or “natural” soil carbon. This is to ask the question, “what is it that makes binding to nanoplastics different in ecological importance?” I think the way forward in publishing this elegant experimental study is to place a more general emphasis on this preliminary study in an interesting perspective on the transport of soil carbon using contrast variation to selectively visualise this soil component.

Response: We thank the reviewer for their comments. First, we agree with the reviewers that environmental pollutants can bind to soils as well as nanoplastics. Moreover, soils can be apolar when hydrophobic molecules adsorb to soil particulates. However, generally, nanoplastics are more nonpolar than soils. For instance, their water-surface contact angles are typically near 90o (e.g., 82o for the PBAT film we used in our study, as is now reported therein). Second, the particle density of nanoplastics is significantly lower, on average, than the particle density for many soils. As a result, settling velocities of nanoplastics will be lower on average than soil particles. 

To address the reviewer’s comments, we have modified the text accordingly (L 89 ff): “.. and differently than soil micro- and nanoparticles (which particularly occur in clays) due to their more hydrophobic nature than most soils. (But, it is noted that adsorption of hydrophobic molecules onto soils can induce hydrophobicity into soils [31].) In addition, the density of NPs for agricultural plastics would significantly lower than soils: 0.5-1 g cm-3 for plastics used in mulch films versus particle densities of 2-3 g cm-3 for many soils [26, 32].”

Reviewer: Specific comments: I think personally that level of detail in the fitting procedure obscures from the simple messages of this work.

Response: We aimed to keep our description of the SANS and USANS analysis to a minimum, but believed that at least some detail was needed, especially given the diversity of the readership of PLOS ONE. But, to address the reviewer’s comments, we greatly shortened the description of form factor-structure factor modeling by omitting Eq 2-3 and citing a general reference. (top paragr of p. 11)

Reviewer: Page 18 line7: Minor typo “NPs were removed from the neutron be through”, neutron beam?

Response: We thank the reviewer for catching this mistake. We have inserted “beam” after “neutron” as recommended (L 262)

Reviewer: Page 18 same paragraph, Ambiguous language:

“These results suggest that convection improved dispersion (increased surface area) of the

NPs exposed to the neutron beam (e.g., via decreasing the agglomeration of NPs) or increased solvent”

Is there some interaction between with the neutron beam which increases dispersion of particles or is something else meant?

Response: We have modified the sentence accordingly: “Ex situ stirring increased the intensity of the SANS signal of PBAT NPs, a result suggesting that convection improved the dispersion of the NPs by disrupting the formation of large agglomerates. An alternate explanation would be that convection increased the extent of solvent penetration into NPs and their agglomerates. The addition of vermiculite reduced the extent of the increase of I(Q).” (L 264-268)

III. Responses to Reviewer 2

Reviewer: Dear Authors,

Thank you for giving me the opportunity to read your article on nanoparticle dispersion. I found the content has an interesting potential. On the other hand, much more clarification and discussion are required to better understand the content and properly evaluate the true value of the work prior to publication. Especially, Materials and Methods should be further clarified. I suggest this article be completely revised before re-submission for the other review processes. As a conclusion, I recommend its major revision at this state.

I hope my comments are helpful.

Good luck,

A reviewer

Major concerns:

Materials and Methods: -“apparent density of …g m-2”-> g m-3 ?

Response: Line 123- For films, It is common to express density as “apparent density” in g m-2 . The density can readily be calculated by dividing the apparent density by the film thickness. To address, we have added the following: “(i.e., a specific gravity of 0.787)” (L126)

Reviewer: -“The 45 μm particle fraction was collected, and an average particle size of 38±12 μm…”->Please provide the justification why this size is appropriate for your study.

Response: To address, we added the following sentences: “Soil particles of this size were selected because of their effective dispersion in water, and their high monodispersity was anticipated to simplify interpretation of the SANS data. Vermiculite particles within the given size range mimic silt [33].” (L 138-141)

Reviewer: -How many particles were counted by ImageJ’s “analyze particles” algorithm?

Response: To address, we added the following sentence: “For each particle size fraction, 250 particles were counted and averaged.” (L 146-147)

Reviewer:-“…1.0wt.% PBAT NPs and/or 0.5% vermiculite…”->How did you select these concentrations? Have you also varies them to investigate their effects? According to your introduction, you wish to mimic low concentration of NPs in soil. It seems the particle concentrations are too low, especially for vermiculite to simulate soil. Your samples are like small number of particles swimming in a plenty of water. In agricultural soil, particle movements/displacement should be limited, with much higher volume of soil particles than the one of water.

Response: The authors agree with the reviewer, that the concentrations were lower than would occur in most soils. However, the conditions evaluated would mimic the interface of soils with groundwater. As stated in the Abstract, Introduction, and throughout the manuscript, this study aims to show a proof of concept, if the agglomeration of nanoplastics can be determined in soil via SANS/USANS using neutron contrasting techniques. The employment of low concentrations, 1wt% NPs and 0.5wt% vermiculite, enabled our hypothesis to be tested. We note that we completed SANS measurements within the last 4 weeks that involve a wider range of NP concentration as well as ex situ stirring times. (The SANS experiments were delayed by the unanticipated 1-year shutdown of ORNL’s High-Flux Isotope Reactor.) We will analyze the data and report on our findings later this year, hopefully. We are investigating other soil mimics that will allow us to achieve a stronger contrast match (i.e., I(Q) achieved being closer to zero), which will allow us to use higher soil concentrations.

Reviewer:- Do you have any more information about the particle properties (e.g. degree of hydrophobicity, particle roughness)? What is your sample environment (e.g. solution pH and salt type/concentration)? Nanoparticle dispersions can be strongly affected by those factors.

Response: To address, we have included 1) the contact angle of the PBAT-based film (82.5±1.1o, indicative of hydrophobicity; L 128; 2) the zeta potential (-22±3.6 mV; L 179); 3) information on the surface roughness (via atomic force microscopy; AFM) and the impact of nanoplastics and vermiculite on pH and conductivity (L 180-176; AFM images provided via the Supporting Information, Fig S1):

“According to Atomic Force Microscopic (AFM) analysis, performed using a model Multimode 8 instrument from Bruker (Santa Barbara, CA, USA), NPs were irregularly shaped and possessed an average roughness of 12.22±1.55 nm (Fig S1 of the Supporting Information).The pH (electrical conductivity) value for the 0.5% vermiculite slurry in water was determined to be 10.14±0.02 (89.57±0.28 µS cm-1) and after the addition of 1% NP to the 0.5% vermiculite slurry to be 9.54±0.13 (80.03±0.29 µS cm-1).”

Reviewer: -How do you justify your ex situ stirring conditions were appropriate/optimized for your study? Stirring conditions should also affect the particle dispersion/aggregation as well as particle concentrations (see my comments above).

Response: The stir rate of 400 rpm was chosen to allow effective dispersion of nanoplastics and artificial soil in solution. The stir rate was not optimized. As commented upon above, we have very recently investigated a wider range of stirring times via SANS and USANS, and will report on the results in the near future. Stir rate will be investigated in future research.

Reviewer:-“…samples were recovered and kept for refrigerated prior to SANS/USANS analysis”->Do you have any proof that this procedure can keep the sample status unchanged? Did you feed your refrigerated sample to a SANS/USANS cell after the sample temperature adjusted to your measurement temperature of 22°C?

Response: Samples after thawing were kept refrigerated at 8°C and adjusted to room temperature 24 hours before measurement. We inserted the following sentence (L 193): “Changes in dp due to refrigeration were within 5% (DLS analysis).”

Reviewer: -“Typical acquisition times were…8-12 h for…USANS…”->Do you have any proof your sample status does not change during such a long period time? Especially, MP samples might sediment during the measurement.

Response: We cannot verify for certain that size reduction was completely absence during the tumbling of the sample in the neutron beam. But, the tumbling rates used for SANS and USANS were very low (10 rpm and 5 rpm for SANS and USANS, respectively). During preliminary experiments, we found that tumbling rates lower than 5-10 rpm did not produce effective dispersion of particles; therefore, the tumbling rates noted above were used.

Reviewer: 2. Results and discussion:

-Figures 1-3: Please add error bars if possible.

Response: We have provided error bars directly to Figs 1B and 3. For Fig 2, error bars are smaller than the symbols; the figure caption was modified to convey this information accordingly. For Fig 1A, the error bars provide too much clutter to the figure. Therefore, we provided the error bars in Fig S2 of the Supporting Information, achieved by dividing Fig A into 2 subfigures. Error bars were also provided for Table 1. An explanation for the error bars of the SANS and USANS data is given in L 211-213.

Reviewer: -Figure 2-A: Please provide the figure legends.

Response: We added the legend to Fig 2A as recommended.

Reviewer: -Figure 2-B: What does “x10, x100, x1000” mean?

Response: As stated in the caption in both versions of this manuscript, “For Fig B, I(Q) was multiplied by a constant (as given in the legend) to improve visualization.”

Reviewer: -“Ex situ stirring increased the intensity of the SANS signal of PBAT NPs…These results suggest that convection improved dispersion of the NPs exposed to the neutron beam…”->Does it mean without stirring particles are aggregated and sedimented? If yes, the sample in the beam without experiencing the stirring is not representative to compare with the sample experienced the stirring. If no, why you did not see the change in the scattering curves? Not only the particle numbers, but also particle/aggregate size should also affect the scattering intensity. Referring to your eq.2, the scattering intensity is proportional to (particle volume)2 while (particle number)1.

Response: We are not certain if we fully understand the reviewer’s questions and comments. First, all SANS and USANS samples underwent slow tumbling (5-10 rpm) to ensure that the suspensions were uniform, as described in our responses above. The authors found, that ex situ stirring increased the dispersion of nanoplastics, which would increase the surface area of nanoplastics exposed to the neutron beam, and therefore increase I(Q).

Reviewer: -“…the attractive forces between NPs must be weak.”->This statement is vague and do not carry any quantitative information. Consider removing it. If you wish to keep it, please provide your proof. See my comments about the particle properties and sample environments.

Response: We removed the sentence as recommended.

Reviewer: 3. Conclusions -“NPs are an emerging threat to soil,…as well as their long-term fate and transport (including to groundwater).”->It sounds like an introduction content. Consider moving and integrating it as a part of introduction.

Response: We believe the sentence should stay in its current position. Moreover, the second paragraph of the Conclusions section provides some perspective of the magnitude of the potential impact of terrestrial nanoplastics and the need for robust approaches to study their fundamental behavior and to detect them in agricultural soils. We believe that SANS+USANS with neutron contrast matching techniques may be particularly valuable to study their fundamental nano-physical chemistry and hope we have demonstrated the method’s potential utility in this paper.

Reviewer: -You may inform possible future work.

Response: We added the following sentence to the end of the Conclusions section: “We are currently evaluating the effect of NP concentration and ex situ stirring time on the agglomeration of NPs and soil by SANS and USANS using the approach described herein” (L 352-354).

Reviewer: Minor concerns:

-Please add line numbers to your manuscript. It makes a reviewer easier to point out his/her concerns.

Response: Line numbers were added 

Reviewer: -Title: “…particulates…”->Consider replacing it with “particles” that is used more in the main text. Please be consistent.

Response: “Particulates” was replaced with “Particles” in the title, as recommended

Reviewer: -Introduction: “…plastic plant-soil interactions…”->Do you mean “plastic-soil interactions”?

Response: Line 60- Thank you for catching this. “Plant” was removed 

Reviewer: -“…recycling and landfilling…is impracticable due to soil contamination.”->Not sure what you mean. Can you consider rephrasing it or explaining more?

Response: Sentence was modified: “..options. Recycling programs are mostly unavailable [20].” (L 67)

Reviewer: -Figure 3: Define “Schulz polydisperse form factor” in the main text.

Response: The text of the Experimental section now reads:” ‘the polydispersity of the radius (based on a Schulz distribution) pd .. [41]” (L229-230). Reference 41 describes the Schultz distribution fully.

---

## [Decision Letter · Decision Letter 1]

17 Jun 2020

PONE-D-19-33737R1

Effects of soil particles and convective transport on dispersion and aggregation of nanoplastics via small-angle neutron scattering (SANS) and ultra SANS (USANS)

PLOS ONE

Dear Dr. Hayes,

Thank you for submitting the revised manuscript. One more comments to be carefully addressed. Can authors respond quickly.  

We look forward to receiving your revised manuscript.

Kind regards,

Pratheep K. Annamalai

Academic Editor

PLOS ONE

Additional Editor Comments (if provided):

Minor revision required, as suggested by the reviewer

Reviewers' comments:

Reviewer's Responses to Questions

**Comments to the Author**

1. If the authors have adequately addressed your comments raised in a previous round of review and you feel that this manuscript is now acceptable for publication, you may indicate that here to bypass the “Comments to the Author” section, enter your conflict of interest statement in the “Confidential to Editor” section, and submit your "Accept" recommendation.

Reviewer #2: All comments have been addressed

2. Is the manuscript technically sound, and do the data support the conclusions?

Reviewer #2: Yes

3. Has the statistical analysis been performed appropriately and rigorously? 

Reviewer #2: Yes

4. Have the authors made all data underlying the findings in their manuscript fully available?

Reviewer #2: Yes

5. Is the manuscript presented in an intelligible fashion and written in standard English?

Reviewer #2: Yes

6. Review Comments to the Author

Reviewer #2: Dear Authors,

Thank you for all the efforts. I found that the quality and clarity of your article have been significantly improved. At this point, I have only one additional comment/suggestion below, based on our first exchange. I hope my comments are helpful.

Best regards,

A reviewer

My comments-“Typical acquisition times were…8-12 h for…USANS…”->Do you have any proof your sample status does not change during such a long period time? Especially, MP samples might sediment during the measurement.

Your Response: We cannot verify for certain that size reduction was completely absence during the tumbling of the sample in the neutron beam. But, the tumbling rates used for SANS and USANS were very low (10 rpm and 5 rpm for SANS and USANS, respectively).

->I was talking about aggregation under the very low agitation during the long USANS measurement. Since you cannot separately evaluate the effect of secondary particle addition on the change in particle/aggregate size and associated scattering intensity from the effect of low agitation, I would suggest that you state this point briefly as an indication of the method limitation and for future improvement, for you and other researchers.

7. PLOS authors have the option to publish the peer review history of their article (what does this mean?). If published, this will include your full peer review and any attached files.

Reviewer #2: No

---

## [Author Response · Author response to Decision Letter 1]

23 Jun 2020

REVIEWER 2: Thank you for all the efforts. I found that the quality and clarity of your article have been significantly improved. At this point, I have only one additional comment/suggestion below, based on our first exchange. I hope my comments are helpful.

Best regards,

A reviewer

My comments-“Typical acquisition times were…8-12 h for…USANS…”->Do you have any proof your sample status does not change during such a long period time? Especially, MP samples might sediment during the measurement.

Your Response: We cannot verify for certain that size reduction was completely absence during the tumbling of the sample in the neutron beam. But, the tumbling rates used for SANS and USANS were very low (10 rpm and 5 rpm for SANS and USANS, respectively).

->I was talking about aggregation under the very low agitation during the long USANS measurement. Since you cannot separately evaluate the effect of secondary particle addition on the change in particle/aggregate size and associated scattering intensity from the effect of low agitation, I would suggest that you state this point briefly as an indication of the method limitation and for future improvement, for you and other researchers.

RESPONSE BY AUTHORS: To address the reviewer’s concern, we added text to the Experimental section:

• L 201-202: “Moreover, the tumbling speed was set to match the settling velocity of particles to ensure that the particles remain mostly in the path of the beam, rather than settle out.”

• L212-215: “We did not observe the settling out of particles at any instance during the SANS or USANS experiments. Although we cannot fully rule out that particle aggregation was induced by low in situ tumbling, the absence of settling gives us confidence that the impact of this event was small.”

---

## [Editor Report · Decision Letter 2]

25 Jun 2020

Effects of soil particles and convective transport on dispersion and aggregation of nanoplastics via small-angle neutron scattering (SANS) and ultra SANS (USANS)

PONE-D-19-33737R2

Dear Dr. Hayes,

Thank you for the revision. We’re pleased to inform you that your manuscript has been judged scientifically suitable for publication and will be formally accepted for publication once it meets all outstanding technical requirements.

Kind regards,

Pratheep K. Annamalai

Academic Editor

PLOS ONE

Additional Editor Comments (optional):

Authors are appreciated for the latest revision addressing all the reviewers' comments.
---

## [Editor Report · Acceptance letter]

8 Jul 2020

PONE-D-19-33737R2 

Effects of soil particles and convective transport on dispersion and aggregation of nanoplastics via small-angle neutron scattering (SANS) and ultra SANS (USANS) 

Dear Dr. Hayes:

I'm pleased to inform you that your manuscript has been deemed suitable for publication in PLOS ONE. Congratulations! Your manuscript is now with our production department. 

Kind regards, 

on behalf of

Dr. Pratheep K. Annamalai 

Academic Editor

PLOS ONE